# Paradoxical Radiosensitizing Effect of Carnosic Acid on B16F10 Metastatic Melanoma Cells: A New Treatment Strategy

**DOI:** 10.3390/antiox11112166

**Published:** 2022-10-31

**Authors:** Miguel Alcaraz, Amparo Olivares, Marina Andreu-Gálvez, Daniel Gyingiri Achel, Ana María Mercado, Miguel Alcaraz-Saura

**Affiliations:** 1Radiology and Physical Medicine Department, School of Medicine, Campus de Excelencia Internacional de Ámbito Regional (CEIR)-Campus Mare Nostrum (CMN), University of Murcia, 30100 Murcia, Spain; 2Applied Radiation Biology Centre, Radiological and Medical Sciences Research Institute, Ghana Atomic Energy Commission, Legon, Accra GE-257-0465, Ghana

**Keywords:** radiation effects, bystander, radioprotectors, radiosensitizers, micronucleus, B16F10, PNT2

## Abstract

Carnosic acid (CA) is a phenolic diterpene characterized by its high antioxidant activity; it is used in industrial, cosmetic, and nutritional applications. We evaluated the radioprotective capacity of CA on cells directly exposed to X-rays and non-irradiated cells that received signals from X-ray treated cells (radiation induced bystander effect, RIBE). The genoprotective capacity was studied by in vivo and in vitro micronucleus assays. Radioprotective capacity was evaluated by clonogenic cell survival, MTT, apoptosis and intracellular glutathione assays comparing radiosensitive cells (human prostate epithelium, PNT2) with radioresistant cells (murine metastatic melanoma, B16F10). CA was found to exhibit a genoprotective capacity in cells exposed to radiation (*p* < 0.001) and in RIBE (*p* < 0.01). In PNT2 cells, considered as normal cells in our study, CA achieved 97% cell survival after exposure to 20 Gy of X-rays, eliminating 67% of radiation-induced cell death (*p* < 0.001), decreasing apoptosis (*p* < 0.001), and increasing the GSH/GSSH ratio (*p* < 0.01). However, the administration of CA to B16F10 cells decreased cell survival by 32%, increased cell death by 200% (*p* < 0.001) compared to irradiated cells, and increased cell death by 100% (*p* < 0.001) in RIBE bystander cells (*p* < 0.01). Furthermore, it increased apoptosis (*p* < 0.001) and decreased the GSH/GSSG ratio (*p* < 0.01), expressing a paradoxical radiosensitizing effect in these cells. Knowing the potential mechanisms of action of substances such as CA could help to create new applications that would protect healthy cells and exclusively damage neoplastic cells, thus presenting a new desirable strategy for cancer patients in need of radiotherapy.

## 1. Introduction

Carnosic acid (CA) (C_2_OH_28_O_4_), present in plant species of the Lamiaceae family (*Rosmarinus officinalis* and *Salvia officinalis*) is a phenolic diterpene. It is classified among the so-called terpenoids, otherwise known as isoprenoids or terpenes. Due to the presence of a phenolic group, CA is often classified among the polyphenols. However, their cellular distribution, biosynthetic pathway, solubility properties, and functions differ substantially from most classes of polyphenols, and closely resemble terpenoids such as tocopherols and carotenoids. It is a fat-soluble compound characterized by high antioxidant capacity, and is used in many industrial applications, such as food, cosmetics, health, and nutrition. Despite its economic potential, surprisingly few studies have been conducted on its absorption, transport, metabolism, and tissue distribution characteristics, which remain poorly understood [1,2,3,4].

CA has been shown to possess important radioprotective capacities against cellular damage induced by ionizing radiation. CA has been found to decrease the genotoxic and cytotoxic effects induced by ionizing radiation [5,6,7,8,9,10] and even ultraviolet radiation [11] when administered on different cell lines. This radioprotective capacity of CA has been attributed to its antioxidant capacity, mediated by its ability to scavenge free radicals induced by the action of ionizing radiation when it interacts directly with irradiated cells (“target” theory) [5,6,7,8,9,10,11]. However, we have not found previous references that describe this radioprotective effect in radiation-induced bystander effect (RIBE) recipient cells or that CA could show a paradoxical radiosensitizing effect in certain cell lines.

Melanoma is known to have the highest mortality rate among skin cancer patients [12,13,14]. Its incidence is markedly high among the white ethnic group. In the United States, the incidence rate of melanoma among the white population is much higher than that found among the Hispanic and the black population (5 and 19 times higher, respectively) [15]. For this reason, it is extremely important to develop new therapeutic arsenals aimed at improving the survival of affected patients [16,17]. Furthermore, it is an aggressive cancer with high resistance to treatment with ionizing radiation and chemotherapy [16,17,18]. The B16F10 melanoma cell line is a mouse metastatic cell line that is frequently used as a research model of melanomas [18,19].

In this work, we evaluated the genoprotective and cytoprotective capacities of CA against lesions induced by ionizing radiation (IR) when it acts directly on irradiated cells (“target” theory) and indirectly on distant cells (radiation-induced bystander effect, (RIBE)). In addition, we specifically examine the radioprotective capacity of CA on normal cells, which are paradoxically transformed into a radiosensitizing effect in melanoma cells. Perhaps the analysis of these opposite effects could help to clarify the mechanisms responsible for the protection of healthy cells while significantly injuring neoplastic cells, thereby achieving a desired treatment strategy for cancer patients undergoing radiation treatment.

## 2. Materials and Methods

### 2.1. Carnosic Acid

Carnosic acid (CA) (C_20_H_28_O_4_) from Rosmarinus officinalis (Figure 1), also known as (4AR,10aS)-5,6-dihydroxy-1,1-dimethyl-7-propan-2-yl-2,3,4,9,10,10a-hexahydrophenanthrene-4a-carboxylic acid, was purchased from Sigma-Aldrich (Madrid, Spain) with a purity of ≥91%.

### 2.2. Antigenototoxic Effect

#### 2.2.1. In Vivo Micronucleus Assay

Swiss male mice of 11 weeks of age were placed in groups with six animals per group. The mice weighed 25–32 g at the beginning of the study. CA was dissolved to a concentration of 1 mg/ ml of CA in DMSO and administered 60 min before x-ray exposure as a single intraperitoneal dose of 0.2 mL. Each study group was made up of six animals: (control (C), treated with CA (CA), irradiated control (Ci), treated with CA and irradiated (CAi), treated with non-irradiated serum (Bys), treated with irradiated serum (Bysi), and treated with irradiated serum and CA (BysCA)). The animals were obtained from and housed in the Laboratory Animal Service of the University of Murcia (REGAES300305440012). All the procedures used in this study were approved by the Research Ethics Committee of the University of Murcia (CECA: 510/2018) and by the Government of the Autonomous Community of the Region of Murcia in Spain (No. A13211208).

Micronucleus Assay in Mouse Bone Marrow (MNPCEs)

The in vivo micronucleus assay on mouse bone marrow was performed according to the technique previously described by Schmid [20]. After 24 h exposure to 50cGy of X-rays, animals were sacrificed by cervical dislocation, bone marrow cells located in the medullary cavity of the femur were extracted, and microscopic preparations were made. Three specialists evaluated the number of micronucleated polychromatic erythrocytes (MNPCE) among 1000 PCE per mouse in a double-blind study. To rule out toxicity induced by CA, the number of normochromatic erythrocytes, total erythrocytes, and the ratio between polychromatophilic and normochromatic erythrocytes were evaluated for each animal.

Flow Cytometry for Micronuclei (MN) in Reticulocytes (MNRET)

Flow cytometry analysis for micronucleated reticulocytes (MNRET) was performed following the technique previously described by Balmus et al. [21]. After 48 h of exposure to 2 Gy of X-rays, blood samples were collected by intracardiac puncture from each animal into blood collecting tubes that contained heparin as anticoagulant. Subsequently, they were fixed in methanol and stored at −80 °C until the beginning of the study. Cells were centrifuged at 500 g for 5 min at 4 °C and the supernatant was decanted followed by the addition of 200 μL of cold bicarbonate buffer and resuspension of pellets by slowly pipetting up and down. Twenty (20) μL of each sample was transferred into wells in 96- deep-well plates (800 μL per well capacity). Eighty (80) μL of CD71-FITC/RNase solution was added to the test samples and incubated for 45–60 min at 4 °C with gentle agitation (~60 r.p.m) on a shaker. The samples (in wells and Eppendorf tubes) were washed by addition of 600 μL of cold bicarbonate buffer (4 °C) and centrifuged at 500 g for 5 min at 4 °C. The supernatant was decanted and the pellets resuspended in 500 μL of bicarbonate buffer. Two hundred microliters (200 μL) of the sample was transferred into clear flat-bottomed 96-well plates and 1 μL PI was added to each well in the plate.

Flow cytometry analysis was performed using a FACS Fortessa with Becton Dickinson high-throughput sampler option (BD Biosciences, San Jose, CA, USA). Approximately 190,000 cells per mouse were analyzed. The frequency of micronucleated reticulocytes (MN-RET) was calculated using the following formula: MN-RET (%) = (MN-RET/RET + MN-RET) × 100.

#### 2.2.2. In Vitro Micronucleus Assay

Micronucleus Test in Vitro (MNCB)

For the in vitro MN assay, we used the cytokinesis-blocked micronucleus assay on human lymphocytes irradiated in vitro using venous blood obtained from the elbow flexure veins of three supposedly healthy young donors. Heparinized blood samples were exposed to X-rays and the cytochalasin B blocked micronucleus (CBMN) assay was performed as described by Fenech and Morley [22] and adapted by the International Atomic Energy Agency to reduce the amount of culture medium used [19]. The samples that were cultured in 4.5 mL of cell culture medium were exposed to X-ray irradiation and the cytochalasin B block micronucleus (CBMN) assay described by Fenech and Morley [22] and adapted by the International Atomic Energy Agency to reduce the amount of culture medium used [23]. In the CA-treated groups, twenty microliters (20 µL) of a 25 µM concentration of CA solution was added to 2 mL of blood. The frequency of MN in cytochalasin-B (CB) blocked cells was determined by analyzing at least 3000 BCs in each of the groups studied and expressing their values in MN/500 BC [18,19]. Slides were digitized with a Leica SCN400F scanner equipped with a Digital Image Hub (Leica Microsystems, Buffalo Grove, IL, USA). The preparations were studied by three specialists (two radiobiologists with more than ten years of experience in cytogenetic studies and a specialist in anatomic pathology (cytopathology) with more than fifteen years of professional practice) to confirm the doubtful images of CBMN. The simples studied were mixed and coded without prior knowledge of the results provided by the other specialists during the analysis process. All the procedures used to handle human samples were approved by the Experimental Biosafety Committee of the University of Murcia (ID: 472/2021). All experiments were repeated six times.

### 2.3. MTT Cytotoxicity Assay

To investigate the cell survival by means of the MTT assay, we used two cell lines with different degree of radioresistance to ionizing radiation [24]: human prostate epithelial cells PNT2, a cell line traditionally considered to be radiosensitive, and the murine metastatic melanoma cell line B16F1, a highly radioresistant cell line. PNT2 cells were obtained from the European Collection of Authenticated Cell Cultures (ECACC) (ECACC 95012613, UK) and were cultured in RPMI-1640 medium supplemented with glutamine (2 mM). B16F10 cells were kindly provided by Dr. Hearing from the National Cancer Institute (Bethesda, MA, USA) and cultured in a 1:1 mixture of Dulbecco’s modified Eagle’s medium (DMEM) and Ham’s F12 (Ham’s Nutrient mixture F-12)). Both mediums were enriched with 10% fetal bovine serum (Gibco BRL) and antibiotics (5% penicillin/streptomycin). Cell cultures were maintained at 37 °C and 95% relative humidity and an atmosphere of 5% CO_2_. Throughout the study, conventional Mycoplasma Tests were conducted to confirm the absence of contamination by Mycoplasma spp. CA dissolved in DMSO (1 mg per mL) was prepared at 25 µM in phosphate buffered saline (PBS). The carnosic acid was administered by adding 25 μL to each well of a concentration of 25 µM CA from this solution. The administration of CA to the culture medium was carried out immediately before X-ray exposure for 15 min. Please see more information in the Appendix A.

#### MTT Assays of Irradiated Cells

To analyze the radioprotective effects of the substances on PNT2 and B16F10 cell lines, two MTT assay types were carried out as previously described [9,10,25,26,27,28,29]. Forty-eight (48) hours after incubation, cell proliferation was determined after exposure to X-radiation. Briefly, PNT2 cells (3200 cells/well) and B16F10 cells (2500 cells/well) were incubated in 200 µL of growth medium and allowed to adhere to the bottom of the wells for 24 h. At 48 h after X-ray exposure, 50 μL of MTT (5 mg/mL) in culture medium was added to each well and incubated in a 5% CO_2_, atmosphere at 37 °C for 4 h. Media and un-metabolized MTT were then removed. After shaking for 30 min at room temperature, the absorbance readings of the plates were read with a Multiskan MCC/340P spectrophotometer using 570 nm for test reading and 690 nm as reference wavelength. Each experiment was repeated six times.

### 2.4. Clonogenic Assay

The clonogenic assay is an in vitro cell survival assay which aptly describes the capability of a single cell to develop into a colony of at least fifty cells. The method essentially tests the ability of each cell in the population to form colonies upon exposure to radiation and to determine the efficacy of CA as a protective or cytotoxic agent. We employed the technique described by Franken et al. (2006) [30]. To do this, we seeded the cells in six-well plates (200 PNT2 cells and 400 B16F10 cells), to which 3 mL of culture medium were added. To allow adhesion, cells were incubated for 24 h after which the test substances were added. Three-hundred (300 µL) microliters of CA at a concentration of 25 µM were added to the CA-treated samples. After 30 min, all samples are irradiated with 4 Gy of X-rays. The samples were incubated for 12 days for colony formation; culture medium was changed every two days. Subsequently, the colonies were fixed in glutaraldehyde (6.0% *v*/*v*) and stained for 30 min with crystal violet (0.5% *w*/*v*). The colonies were washed in copious amount of water and allowed to dry. Colonies were counted using a stereoscopic microscope. The number of colonies counted after treatment of the cells were expressed in terms of PE (ratio of the number of colonies observed to the number of cells seeded) and survival fraction (SF). Where SF = nº of colonies formed after treatment/ nº of cells seeded x PE; and PE = nº of colonies formed / nº of cells seeded. All experiments were repeated six times.

### 2.5. Anexin V

For cell apoptosis determination by flow cytometry, we used the Alexa Fluor^®^ 488 annexin V/Dead Cell Apoptosis Kit (Catalog nos. V13241) (Invitrogen™, Thermo Fisher Scientific, Spain, Madrid), which allows measurement of early apoptosis by detecting the expression of phosphatidylserine (PS) and membrane permeability. The procedure was carried out according to the manufacturer’s instructions. Treated cells were washed once with cold phosphate buffered saline (PBS) and harvested. After centrifuging and resuspending the cells, 5 μL of Alexa Fluor^®^ 488 annexin V was added (component A) followed by the addition of 1 μL of 100 μg/mL PI to each 100 μL of cell suspension. After 15 min incubation, 400 µL of buffer was added to the samples on ice. Stained cells were immediately analyzed by flow cytometry, measuring fluorescence at 530 nm and 575 nm using 488 nm excitation in FACSCalibur (Becton Dickinson, Franklin Lakes, NJ, USA).

### 2.6. GSH Assay

The GSH/GSSG-GloTM assay (Promega, Madison, MI, USA) was used to determine and quantify the levels of total glutathione (GSH + GSSG), reduced glutathione (GSSG), and the GSH/GSSG ratio in PNT2 cells and B16F10 subjected to the different experimental conditions three hours after exposure to 20 Gy of X-rays. The method was carried out according to the manufacturer’s instructions. Two hundred microliters (200µL) of complete medium and cells were seeded into each of the wells of 96-well microtiter plate and allowed to adhere to the base of the wells. The cells were then treated with 25 µL CA solution. Cell densities were evaluated and corrections made using the Bradford assay [31]. Cells were dislodged from the substratum by trypsinization and their fluorescence intensity analyzed using FLUOstar^®^ Omega (BMG Labtech, Offenburg, Germany). All experiments were repeated six times.

### 2.7. Models of Radiation-Induced Bystander Effects (RIBE)

For the evaluation of RIBE, we used the procedure previously described by Olivares et al. [32]. Briefly, for the in vivo micronucleus assay, 0.2 mL of serum collected from control or previously irradiated animals (50 cGy and 2 Gy) were administered intraperitoneally to the test animals. The CA Bys test group also received 0.2 mL of a CA/DMSO solution (1 mg CA/1 mL DMSO) in PBS administered in a single dose by intraperitoneal injection 15 min before the injection of the serum from the irradiated animals (Figure 2a). The determination of the frequency of MN appearance in polychromatic erythroblasts (MNPCEs) and in reticulocytes (MNRET) was performed following the procedure previously described in this study.

To produce the RIBE effect in the in vitro cytokinesis-blocked micronucleus (CBMN) assay, 0.1 mL of serum from irradiated blood samples was added to the blood samples before initiation of lymphocyte culture. Additionally, in the BysCA group, 25 microliters (µL) of a 25 micromolar (µM) solution of CA/DMSO was added (Figure 2b). All experiments were repeated six times.

In the MTT assays, we used the “medium transfer” protocol from irradiated cells to non-irradiated cells. For this protocol, PNT2 and B16F10 cells were plated in T25 cm^2^ flasks at confluency. Before irradiation, the medium was changed for fresh medium in all cells (non-irradiated cells (C), irradiated cells (Ci), and CA irradiated cells (CAi)); this medium was collected 4 h after irradiation to allow the bystander factors to be expressed. The conditioned medium was then centrifuged (200 g) and transferred into microplates containing confluent PNT2 and B16F10 cells. In the groups treated with CA (BysCA), 25 µL of a 25 µM solution of CA/DMSO in PBS was added. The cells were incubated for 48 h and analyzed following the conditions previously described in the MTT and clonogenic assays on cells directly exposed to X-rays (Figure 2c). All experiments were repeated six times.

### 2.8. Irradiation

The exposure to different doses of ionizing radiation was carried out using an Andrex SMART 200E X-ray generator (Yxlon International, Hamburg, Germany) operating with the same technical characteristics (200 kV, 4.5 mA, 2.5 mm Al filtration, 35 cm focus-object distance (FOD) and a dose rate of 1.3 cGy/s). The modification of the exposure time determines the total dose of X-rays administered to the animals and cell cultures in each of the tests carried out. For the in vivo micronucleus assay, the animals were total body irradiated to a single exposure of 50 cGy for MNPCE in bone marrow evaluation and 2 Gy for the determination of MNRET by flow cytometry. For in vitro genotoxicity, whole human blood samples were exposed to 2 Gy X-rays for the determination of micronuclei in cells blocked with cytochalasin B (MNCB). In the MTT assay and GSH assay, cell cultures grown in microplates were irradiated to 20 Gy. In the clonogenic assay, cell cultures were irradiated to 4 Gy. The administered radiation doses were continuously monitored inside the X-ray cabinet and the final radiation dose was confirmed by means of thermoluminescent dosimeters (GR-200^®^; Conqueror Electronics Technology Co Ltd., Shenzhen, China).

### 2.9. Statistical Analysis

In the in vivo and in vitro micronucleus, apoptosis, and intracellular glutathione determinations, analysis of variance supplemented with mean contrasts was used to assess the degree of correlation between variables. Quantitative means were compared by regression and linear correlation analysis. To evaluate the magnitude of protection or protection factor (PF) with respect to the reduction in MN frequency, the following formula [33] was used: PF (%) = (Firradiated control−Firradiated treated/Firradiated control) × 100, where F is the frequency of MN determined in each of the groups studied.

In the cell survival assay, an analysis of variance (ANOVA). of repeated means was performed, and was complemented by the least significant difference (LSD) test. In this case, we modified the previous formula [33] to adapt the PF to cell survival: PF (%) = (Mirradiated control−Mirradiated treated/Mirradiated control) × 100, where M is the percentage of mortality with respect to non-irradiated control cells.

## 3. Results

### 3.1. Antigenototoxic Effect 

#### 3.1.1. In Vivo Micronucleus Assay

Micronucleus Assay in Mouse Bone Marrow (MNPCEs)

The basal micronuclei frequency determined in the control animals (C) and in those treated with CA were similar, and no statistically significant differences were observed, which expresses the absence of a genotoxic effect of CA. For the irradiated control group (Ci), exposure to 50 cGy of X-rays produced an increase in the frequency of appearance of MN compared to control animals (C), showing a significant difference (*p* < 0.001) between both groups, which expresses genotoxic damage induced by X-rays. However, when CA is administered to the animals before X-ray exposure (CAi), there is a significant (*p* < 0.001) reduction in micronuclei yield compared to irradiated animals (Ci). The group treated with CA and irradiated (CAi) shows a Protection Factor (PF) of 79.9 ± 3.4%, which expresses its genoprotective capacity against damage induced by X-rays (Figure 3).

In the animals treated with 25 microliters of serum from non-irradiated animals (Bys), no significant differences were determined with respect to the control animals and those treated with CA, expressing the absence of a genotoxic effect induced by the administered intraperitoneal serum. However, the intraperitoneal administration of serum from animals irradiated with X-rays (Bysi) produced a significant increase in the appearance of micronuclei (*p* < 0.01) compared to the group treated with serum from non-irradiated animals (Bys), which expresses a genotoxic capacity of the serum from the irradiated animals. Finally, intraperitoneal administration of CA before administration of serum from irradiated animals. Furthermore, (Bys CA) shows a reduction in the frequency of MN compared to irradiated and untreated animals (Bysi) (*p* < 0.01), which expresses a genoprotective capacity of CA in these animals with a Protection Factor (PF) of 47.8 ± 2.3%. From our experiments in PCE cells, our obtained results portray the following relationship in the frequency of MN appearance in ascending order: C ≈ CA ≈ Bys < CAi < BysCA < Bysi < Ci (*p* < 0.001)) (Figure 3).

Flow Cytometry for Micronuclei (MN) in Reticulocytes (RET)

The basal micronuclei frequency determined in the control animals (C) and in those treated with CA (CA) were similar, and no statistically significant differences were observed, which expresses the absence of a genotoxic effect of CA. Exposure of animals to 2 Gy of X-rays (Ci) produced an increase in the frequency of appearance of MN compared to control animals (C), expressing a significant difference (*p* < 0.001) between both groups, which in turn expresses an induction of x-ray induced genotoxic damage. The administration of CA before exposure to X-rays (CAi) showed a significant reduction (*p* < 0.001) in micronuclei yield compared to irradiated animals (Ci), which expresses its ability to protect against genotoxic damage induced by ionizing radiation. The group treated with CA and irradiated (CAi) showed a PF of 73.2 ± 3.12%, which expresses the genoprotective capacity of CA against damage induced by X-rays (Figure 4).

In the animals treated with 25 microliters of serum from non-irradiated animals (Bys), no significant differences were determined with respect to control animals (C) and those treated with CA (CA), expressing the absence of genotoxic effect induced by serum when administered intraperitoneally. However, the intraperitoneal administration of serum from animals irradiated with X-rays (Bysi) produced a significant increase (*p* < 0.01) in the appearance of micronuclei compared to the group treated with serum from non-irradiated animals (Bys), which expresses the genotoxic capacity of the serum from the irradiated animals. Finally, the intraperitoneal administration of CA before the administration of serum from irradiated animals (BysCA) shows a reduction in the frequency of MN compared to irradiated and untreated animals (Bysi) (*p* < 0.01), which expresses the genoprotective capacity of CA in these animals, with a PF of 48.48 ± 2.4%. Our results established the following relationship in the frequency of MN appearance in reticulocytes, in ascending order: C ≈ CA ≈ Bys < CAi < BysCA < Bysi < Ci (*p* < 0.001) (Figure 4).

#### 3.1.2. In Vitro Micronucleus Assay

Micronucleus test in vitro (CBMN)

The basal micronuclei frequency established in control animals (C), in those treated with CA, and in animals treated intraperitoneally with 25 microliters of serum from non-irradiated animals (Bys) were similar, and no statistically significant differences were observed, which shows the absence of genotoxic effect of CA. Exposure of animals to 2 Gy of X-rays (Ci) produced an increase in the frequency of appearance of MN compared to control animals (C), portraying a significant difference (*p* < 0.001) in micronuclei yield among both groups, which expresses the genotoxic damage induced by X-rays. The administration of CA before exposure to X-rays (CAi) showed a significant reduction (*p* < 0.001), in micronuclei yield compared to irradiated animals (Ci). The group treated with CA and irradiated (CAi) showed a PF of 41.9 ± 3.12%, which expresses its genoprotective capacity against X-ray-induced damage (Figure 5).

The intraperitoneal administration of serum from animals irradiated with X-rays (Bysi) produced a significant increase (*p* < 0.01) in the appearance of micronuclei compared to the group treated with serum from non-irradiated animals (Bys), which expresses the genotoxic ability of serum from irradiated animals. However, the intraperitoneal administration of CA before the administration of serum from irradiated animals (BysCA) produced a significant reduction (*p* < 0.01) in the frequency of MN expressed compared to irradiated and untreated animals (Bysi). This expresses the genoprotective capacity of CA in these animals, with a PF of 35.5 ± 2.5%. Our experiments with the cytochalasin B block micronucleus assay (CBMN) yielded results with frequency of appearance of MN in the following ascending order: C ≈ Bys ≈ CA < CAi ≈ BysCA ≈ CAi < Bysi < Ci (*p* < 0.001) (Figure 5).

### 3.2. Cytotoxicity Assay. MTT Assay

PNT2 Cells

The administration of CA or other medium obtained from non-irradiated PNT2 cells (Bys) did not modify cell survival compared to the survival determined in control PNT2 cells (C), showing the absence of cytotoxicity of CA at the concentration tested when the cells were incubated for 48 h. Therefore, the representation of the control group (C) in Figure 6 represents these other two groups (CA and Bys) as well.

In the PNT2 cells, exposure to 20 Gy of X-rays (Ci) produced a significant decrease in cell survival, which was a statistically significant difference (*p* < 0.001), portraying the cytotoxic capacity of X-rays. The administration of CA to PNT2 cell cultures before irradiation (CAi) produced a significantly different increase in cell survival (*p* < 0.001) with respect to the Ci group, which shows the cytoprotective capacity of CA against cytotoxic damage induced by X-rays. A PF of 97 ± 1.1% was established in these cells, which expresses the radioprotective capacity of CA, eliminating 67% of radiation-induced cell death (*p* < 0.001).

In the PNT2 recipient cells of the RIBE effect, the group treated with cell medium from cell cultures irradiated with 2 Gy (Bysi) showed a significant decrease in cell survival (*p* < 0.01) compared to control PNT2 cells (C), which shows cytotoxicity induced by the irradiated culture medium. However, the administration of CA on these cells (BysCA) did not show significant differences in cell survival, expressing a lack of effect of CA in these Bys (Bysi) cell cultures. Therefore, in PNT2 cells, the order established in descending order of cell survival was C ≈ CA ≈ CAi > BysCA ≈ Bysi > Ci (*p* < 0.001) (Figure 6).

B16F10 Cells

The administration of CA (CA) or medium obtained from non-irradiated B16F10 cell cultures (Bys) did not modify cell survival compared to the survival determined in control B16F10 cells (C), showing the absence of cytotoxicity of CA at the concentrations tested in the cells when they were incubated for 48 h. Therefore, in Figure 6 the representation of the control group (C) represents these other two groups (CA and Bys) as well.

In the B16F10 cells, exposure to 20 Gy of X-rays (Ci) produced a significant decrease in cell survival, which was determined to be a statistically significant differences (*p* < 0.001), showing the cytotoxic capacity of X-rays. However, contrary to expectations, the administration of CA to B16F10 cell cultures before irradiation (CAi) produced a decrease in cell survival that was shown to be a statistically significant difference (*p* < 0.001), expressing an increase cytotoxic damage induced by X-rays in the irradiated control group (Ci). No protection factor was determined in these cells, which expresses the absence of protective capacity of CA for these cells. On the contrary, a decrease in cell survival of 32.1 ± 3.5% was determined, expressing a radiosensitizing capacity of CA on B16F10 cells, increasing cell death by two times compared to irradiated cells (*p* < 0.001) (Figure 6).

In the B16F10 recipient cells of the RIBE, the group treated with cell medium that came from cell cultures irradiated with 2 Gy (Bysi), showed a significant decrease in cell survival (*p* < 0.01) compared to control B16F10 cells (C), which shows the cytotoxic capacity induced by the irradiated culture medium. The administration of CA to these cells before the addition of irradiated cell culture medium (BysCA) showed a decrease in cell survival (*p* < 0.05) compared to the Bysi group, expressing an increase in cytotoxic damage induced by X-radiation. No protection factor was established in these cells. On the contrary, a decrease in cell survival of 100.1 ± 3.5% was established with respect to the group treated with irradiated serum (Bysi) which expresses the radiosensitizing capacity of CA on B16F10 cells (Figure 6). Therefore, in B16F10 cells, in descending order of cell survival, the following order was established: C > Bysi > BysCA ≈ Ci > CAi (*p* < 0.001) (Figure 6).

### 3.3. Clonogenic Assay

The administration of CA (CA) or medium obtained from non-irradiated cell cultures (Bys) did not modify the survival fraction (SF) compared to the SF determined in control cells (C), showing the absence of cytotoxicity of CA at the concentrations tested. Therefore, in Figure 7 the representation of the control group (C) represents these other two groups (CA and Bys) as well.

In PNT2 cells, exposure to 4 Gy of X-rays (Ci) produced a decrease in SF, which was determined to be a statistically significant difference (*p* < 0.001) compared to the non-irradiated control group (C), showing the cytotoxic capacity of X-rays. The administration of CA to PNT2 cell cultures before irradiation (CiCA) produced an increase in SF compared to the irradiated control group (Ci) established to be a statistically significant difference (*p* < 0.01), showing the cytoprotective capacity of CA against radiation-induced cytotoxic damage. In these cells, a PF of 24.4 ± 2.1% was determined, expressing the radioprotective capacity of CA (Figure 7).

The SF of PNT2 cells treated with clarified cell culture medium obtained from PNT2 cell cultures irradiated to 2 Gy (Bysi) decreased (*p* < 0.01) relative to control PNT2 cells (C), which shows the cytotoxicity induced by the irradiated cell culture medium. However, the administration of CA on these cells (BysCA) did not show significant differences in cell survival, expressing a lack of effect of CA on SF in these Bys cells (Bysi). These two groups (Bysi and BysCA) show significant differences with respect to irradiated control cells (Ci) (*p* < 0.01), portraying higher survival in these cells than in cells directly exposed to X-rays (Ci). Therefore, in PNT2 cells, from highest to lowest cell survival, the following order was established: C ≈ CA ≈ Bys > Bysi ≈ BysCA > CiCA > Ci (*p* < 0.001) (Figure 7).

The administration of CA did not modify the survival fraction (SF) with respect to what was determined for the control B16F10 cells (C), showing the absence of cytotoxicity of CA at the concentrations tested. In B16F10 cells, exposure to 4 Gy of X-rays (Ci) produced a decrease in SF that was determined to be a statistically significant difference (*p* < 0.001), showing X-ray induced cytotoxicity. The administration of CA to B16F10 cell cultures before irradiation (Ci) produced a significant decrease (*p* < 0.01) in SF with respect to the SF determined for irradiated B16F10 cells (CiCA), showing an increase in cytotoxic capacity of CA in these cells. A PF was not determined in these cells; on the contrary, a decrease in SF of over 60% ± 5.3% was determined in irradiated cells (Ci), along with a radiosensitizing and damage enhancement factor 32.7 ± 3.1% higher than the expected damaging effect from X-rays (Ci). All these express the radiosensitizing capacity of CA on B16F10 cells (Figure 7).

The SF of B16F10 cells treated with clarified cell culture medium from B16F10 cell cultures irradiated with 2 Gy (Bysi) decreased compared to control B16F10 cells (C) (*p* < 0.01), showing the cytotoxicity induced by the irradiated culture medium. Under these circumstances, the administration of CA to these cells (BysCA) did not produce a decrease in SF with respect to cells treated with irradiated medium without CA (Bysi), expressing a lack of toxic effect of CA in these control cell cultures. These two groups (Bysi and BysCA) show significant differences with respect to irradiated control cells (Ci) (*p* < 0,01), portraying higher survival in these cells than in cells directly exposed to X-rays (Ci). CA shows no toxicity in these cells, which were not directly exposed to X-rays and in which the RIBE effect (BysCA) is shown, as determined in the control treated with CA and in the irradiated group (CiCA). All these observations express the loss of protective or sensitizing capacity of CA on recipient B16F10 cells (Figure 7). Therefore, in B16F10 cells, the following SF in descending order was established: C ≈ CA ≈ Bys ≈ Bysi > BysCA > Ci > CiCA (*p* < 0.001) (Figure 7).

### 3.4. Apoptosis

The apoptotic investigations showed statistically significant differences between the two studied cell lines. In the PNT2 cells, the administration of CA produced a significant decrease in the percentage of apoptotic cells (*p* < 0.01) compared to the control cells (C), demonstrating the ability of CA to decrease apoptosis in these cells. Exposure of the control cells to 20 Gy of X-rays (Ci) produced a significant increase in apoptotic cells (*p* < 0.001), which expresses the level of cellular damage induced by direct exposure to IR. The results show that the PNT2 cells treated with CA and radiation (CiCA) produced a significant reduction in apoptotic cells with respect to what was determined in cells treated with radiation only (Ci) (*p* < 0.001). This expresses the radioprotective capacity of CA against damage induced by X-rays, practically placing the percentage of apoptotic cells at the level obtained by the administration of CA in non-irradiated cells (CA).

In addition, the administration of the irradiated culture medium obtained from irradiated PNT2 cells (Bysi) cultures showed a significant increase in apoptosis (*p* < 0.01) compared to the control cells (C), expressing an induced bystander effect in the recipient cells. Similarly, the administration of CA in these cells produced a decrease in the percentage of apoptotic cells (*p* < 0.01). The following order in the percentage of apoptotic cells was observed, from lowest to highest: CA ≈ CiCA ≈ BysCA < C < Bysi < Ci (*p* < 0.001) (Figure 8).

In the B16F10 cells, the administration of CA did not produce significant changes in the percentage of cells undergoing apoptosis with respect to that determined in control cells (C), expressing the lack of capacity of CA to decrease apoptosis in these cells. Exposure of the irradiated control cells (Ci) to 20 Gy of X-rays yielded a significant increase in apoptotic cells (*p* < 0.001), which expresses the cell damage induced by direct exposure to ionizing radiation. B16F10 cells treated with both CA and radiation (CiCA) showed a significant reduction in apoptotic cells with respect to that pertaining in cells that only received radiation (Ci) (*p* < 0.01). However, it can be noted that the percentage of apoptotic cells is still five times higher than what was determined in control cells (C) or unirradiated cells treated with CA (CiCA) (*p* < 0.001) (Figure 8).

The administration of irradiated culture medium obtained from irradiated B16F10 cells (Bysi) cultures showed a significant increase in apoptosis (*p* < 0.01) compared to control cells (C), expressing an induced bystander effect in the recipient cells. However, the administration of irradiated serum previously treated with CA (BysCA) did not show a significant difference in the percentage of the apoptotic cells compared to the group not treated with CA (Bysi). The following order was observed from the lowest to the highest percentage of apoptotic cells: C ≈ CA < Bysi ≈ BysCA < CiCA < Ci (*p* < 0.001) (Figure 8).

### 3.5. GSH Assay

The evaluation of total cellular GSH content showed statistically significant differences between the studied two cell lines. B16F10 cells presented a much higher amount of total GSH than PNT2 cells (*p* < 0.001), which was practically double the concentration of total GSH determined in PNT2 cells (B16F10 > PNT2, *p* < 0.001). In the PNT2 cells, significant differences were observed between the control cells (C) and the other groups of cells studied, demonstrating that all these groups (CA, Ci, CiCA, Bysi and BysCA) show a significant decrease in total glutathione levels compared to the control cells (C) (*p* < 0.001). In the PNT2 cells, the following descending order of total Glutathione concentration was established: C > CA ≈ Ci ≈ CiCA ≈ Bysi ≈ BysCA (*p* < 0.001). In B16F10 cells, the differences observed between the different groups were not statistically significant (Figure 9).

In the PNT2 cells, the administration of CA produced a significant increase in the GSH/GSSH ratio (*p* < 0.01), which expresses the ability of CA to increase the GSH/GSSH ratio in the treated cells, thereby reducing the GSH concentration increase in the cytoplasm of these cells. Exposure of cells to 20 Gy of X-rays (Ci) produced a significant decrease in the GSH/GSSH ratio during the period studied (*p* < 0.001), which expresses the decrease in concentration of reduced glutathione as a consequence of direct exposure to ionizing radiation. The results show that PNT2 cells treated with CA and irradiation (CiCA) showed a lower GSH/GSSH ratio than that determined in non-irradiated cells (C) (*p* < 0.01), which expressed the damage caused by X-rays. In addition, the increase in the GSH/GSSH ratio with respect to irradiated cells (Ci) (*p* < 0.05) expresses the protective capacity of CA (Figure 9).

The administration of irradiated culture medium obtained from irradiated PNT2 cell (Bysi) cultures showed a reduction in the GSH/GSSH ratio with respect to the control cells (C) (*p* < 0.01), and portrays the lesions produced by the administration of the irradiated medium to the receptor cells. However, when the administration of the irradiated medium is preceded by the administration of CA (BysCA) we did not observe significant differences in the GSH/GSSH ratio compared to cells not treated with CA (Bys), showing the absence of any protective capacity of CA for these cells where RIBE was induced. From the highest to lowest ratio of GSH/GSSH, our results can be expressed in the following order: CA > C > CiCA ≈ Bysi ≈ BysCA > Ci (*p* < 0.001) (Figure 10).

In the B16F10 cells, the GSH/GSSH ratios in controls (C), in cells treated with CA, and in irradiated cells (Bysi) were similar, and no statistically significant differences could be deduced between them. Exposure of control cells (Ci) to 20 Gy of X-rays produced a significant decrease in the GSH/GSSH ratio during the study period (*p* < 0.001), which expresses a decrease in reduced glutathione concentration as a consequence of direct exposure to ionizing radiation. The results show that the B16F10 cells treated with CA and radiation (CiCA) showed a GSH/GSSH ratio lower than what was determined in the non-irradiated cells (C) (*p* < 0.001). This observation expresses the damage caused to the cells by X-rays, while presenting an even greater decrease in the GSH/GSSH ratio with respect to irradiated cells (Ci) (*p*< 0.05); the latter expresses the greater sensitization of these cells to radiation upon the administration of CA. The administration of irradiated culture medium obtained from irradiated B16F10 cell cultures (Bysi) did not show any significant differences in GSH/GSSH ratio compared to the control cells (C). However, the administration of CA to these cells (BysCA) produces a significant increase in the GSH/GSSH ratio compared to the control group (C) (*p* < 0.01) and the Bysi group (*p* < 0.01). From highest to lowest GSH/GSSH ratios, our results can be expressed in the following order: BysCA > C ≈ CA ≈ Bysi > Ci > CiCA (*p* < 0.001) (Figure 10).

## 4. Discussion

The micronucleus (MN) test is one of the most widely used methods employed to assess chromosomal damage caused by chemical and physical agents [34,35,36], and its usefulness for characterizing genotoxicity has been endorsed by many different international organizations (IAEA, OECD, OIN) [37]. The in vitro MN assay on irradiated human lymphocytes with cytokinesis blocking (CBMN) is a widely validated technique that allows for the evaluation of DNA instability induced by genotoxic agents, and is especially recommended in the search for genoprotective substances [22,38]. In this assay, MN is defined as a chromosomal fragment that remains outside the spindle during mitosis and remains isolated in the cytoplasm after cell division.

On the other hand, the in vivo MN assays allows the evaluation of other complex biological parameters which complement the expected results. The rodent bone marrow MN test is the most widely used MN test for the determination of genotoxicity in vivo [20,21]. In the in vivo studies in mammals, an MN is defined as a DNA fragment that remains in the cytoplasm when the main nucleus has already been expelled during erythropoiesis. MN are easily recognizable as rounded structures due to their chromatin staining characteristics, which are similar in intensity and texture to the main nucleus located in the cytoplasm [39,40]. The addition of flow cytometric techniques for the determination of reticulocytes in blood represents the latest modification to the in vivo MN assay. This allows confirmation of results and reduction in statistical biases due to the evaluation of hundreds of thousands of cells in each assay [21].

Ultimately, MNs reflect chromosomal aberrations caused by chromosomal breaks induced by errors during DNA replication that are seen in the cell cytoplasm after cell division. Although a basal or spontaneous frequency of MN can be determined in normal populations, this frequency increases significantly after exposure to genotoxic agents or mutagenic substances [34,35,36,37,38,39,40,41]. There is no single assay recommended for genotoxicity evaluation. Therefore, the interrelation of the results obtained between the three tests previously described allows for a more accurate evaluation of the genotoxic effects of any agent or substance tested.

We used these three different cytogenetic assays in a complementary way to confirm the genotoxic capacity of X-rays by determining the frequency of appearance of MN in irradiated human lymphocytes and in animals exposed to ionizing radiation. On the other hand, a reduction of this MN frequency confirms the protective capacity (antigenotoxic effect) of CA against damage induced by ionizing radiation [38,39,40,41]. In addition, it was possible to determine the in vivo and in vitro genotoxic effects of irradiated serum from recipient animals and irradiated medium obtained from irradiated cultures cells, respectively, in order to characterize the RIBE [32].

References on the effects CA against genotoxicity produced by the radiation induced bystander effect (RIBE) are not available. Our results show a significant reduction in the frequency of appearance of MN, which expresses a genoprotective capacity against this induced genotoxic damage in recipient cells. The homogeneity of results obtained in this study permitted us to determine the genoprotective capacity of CA in vivo and in vitro. This highlights the genoprotective effect shown in cells directly exposed to radiation and in recipient bystander cells (RIBE).

The inhibition of cell growth is one of the parameters used to assess the cytotoxic effect of chemical substances and physical agents. In radiobiology, this goes by the name of Cell Survival [24]. In this study, we employed two assays widely used for the assessment of cell survival: the MTT assay [9,10,28,29] and the in vitro clonogenic assay. Different studies have described the cytotoxic effects of ionizing radiation on PNT2 cells [9,10,28,29]. Previous references on the effect of CA on cell survival in PNT2 cells exposed to IR are unavailable. In our study, as was expected, when exposed to 20 Gy of X-rays, CA showed a significant increase in cell survival, which expresses its high radioprotective capacity. This radioprotective capacity is similar to that described for other substances, and has been attributed to its antioxidant property, which confers the ability to eliminate free radicals induced by ionizing radiation [9,10,28,29].

Previous studies have described the cytotoxic effect of ionizing radiation on B16F10 cells [9,10,29]. However, in the B16F10 cells, contrary to the results that were expected, the administration of CA produced a significant decrease in cell survival, demonstrating a paradoxical radiosensitizing capacity that does not correlate with the radioprotective capacities of CA in other lines or cell assays [9,10,28,29]. Previous studies of the effect of CA on cell survival in irradiated B16F10 cells are unavailable. However, different polyphenolic compounds and diterpenes have shown antiproliferative capacity, inhibiting cell growth in B16F10 cells [25,26,41,42] and decreasing metastatic invasion in vivo [43,44]. This inhibition of cell growth is increased if these substances are administered before exposure to ionizing radiation [9,10]. Different authors have even demonstrated an antiproliferative capacity of CA in combination with different antitumor drugs on B16F10 cells [43,44,45,46,47], showing inhibition of adhesion and metastatic migration of melanoma cells, possibly due to inhibition of epithelial-mesenchymal transition (EMT) and inactivation of AKT [43,45,46].

Considering that many authors have related the radiosensitivity or radioresistance of cells with glutathione [47,48,49,50,51,52,53], we proposed to evaluate the concentration of glutathione in both cell lines studied. Together, glutathione and its precursor cysteine (Cys) are the main intracellular antioxidants with the ability to effectively quench ROS induced by IR and reduce oxidative stress [54,55]. In our study, an important difference between the two studied cell types is the unique capacity of melanocytes to produce melanin. In melanocytes, Cys has an additional fate due to its involvement with one of the melanogenesis pathways to form pheomelanin [54]. Activation of the pheomelanin synthetic pathway or an increase in the intracellular concentration of Cys and/or GSH levels increases their use in pheomelanin synthesis, to the detriment of their availability for other cellular activities [49,50,52].

Our results show that B16F10 cells have a much higher total glutathione concentration than PNT2 cells, confirming the previously described results [48]. Numerous authors have attributed the resistance of cells to treatment with chemotherapeutic substances and ionizing radiation to the amount of intracellular glutathione [56,57,58,59,60]. This large difference in the amount of total intracellular glutathione in B16F10 cells could explain the greater radioresistance of these cells in our study.

Previous references on the effect of CA on glutathione concentration in PNT2 cells exposed to ionizing radiation are not available. Our results show that CA administration produces a decrease in the amount of total GSH concentration in all the group of cells (Ci, CiCA, Bysi and BysCA) compared to the control group (C). This could express the cytotoxic capacity of the cells which received CA, exposure to X-rays, and irradiated culture medium. However, a significant increase in the GSH/GSSH ratio was determined in both irradiated and non-irradiated cells, which could explain part of the radioprotective capacity of CA in PNT2 cells. This increase in the proportion of reduced glutathione could explain part of the radioprotective capacity of CA in PNT2 cells. Our results support the hypothesis described by other authors about the radioprotective effect of an intracellular antioxidant, as it acts in a complementary or additive way with intracellular endogenous glutathione, thus helping to maintain high levels of reduced intracellular glutathione [9,10,59,60].

Previous references on the effect of CA administration on intracellular GSH in B16F10 cells are not available. In our study, the administration of CA to B16F10 cells did not modify the amount of total glutathione or the GSH/GSSG ratio. However, after X-ray exposure, CA administration resulted in a significant decrease in the GSH/GSSG ratio. This implies a reduction in the reduced glutathione available by the cell to eliminate the free radicals induced by IR, which in general terms is related to the increase in oxidative stress induced by IR, and can be interpreted as a sign of the damage induced by IR [48]. 

It has previously been described that CA can increase apoptosis in different cell lines by inducing the expression of caspases 3, 8, and 9 and by affecting the Akt/mTOR pathway [47,61,62]. In our study, we did not find differences in the percentage of apoptotic cells after administration of CA to the PNT2 and B16F10 cell lines. However, as irradiated cells have obviously been subjected to oxidative stress, we established a reduction in the percentage of apoptotic cells induced by radiation in both cell lines, although with different degrees of intensity. In this sense, it has been contemplated that the depletion of intracellular GSH might be an early hallmark in the progression of cell death in response to different apoptotic stimuli [63,64,65,66]. Depletion of GSH levels induces and/or stimulates apoptosis, while high levels of intracellular GSH have been associated with decreased apoptosis in different models [63,64,65,66]. In our study, the observed increase in the GSH/GSSG ratio in PNT2 cells implies an increase in reduced GSH levels, which could reduce the percentage of expected apoptotic cells by exposure to ionizing radiation. In contrast, in B16F10 cells, the decrease in the GSH/GSSG ratio observed in cells treated with CA and exposed to radiation implies a reduction in reduced GSH, which although slightly decreasing the percentage of apoptotic cells, does not compensate for the damaging effect induced by ionizing radiation.

In our study, the paradoxical radiosensitizing effect mediated by the decrease in the GSG/GSSG ratio produced by CA in B16F10 melanoma cells could be due to the sum of different factors. CA could, as previously described for caffeic acid [67] and rosmarinic acid [10,48], induce the activation of melanogenesis through the pheomelanin pathway and cause a decrease in intracellular levels of GSH. As this is used for the production of this pheomelanin, it therefore cannot be directed to eliminate ROS induced by IR. Furthermore, the decrease in the GSH/GSSG ratio at reduced GSH levels could be due to the lower activity of superoxide dismutase described in the B16F10 cell line [46,67]. It could even be due to a possible inhibitory effect of CA on glucose-6-phosphate dehydrogenase, glutathione reductase [68], or glutathione S-transferase, as has been described in diterpenes obtained from the Lamiaceae family (rosmarinic acid and carnosol) [69], which decrease cell capacity to reduce oxidized glutathione by reducing intracellular concentrations of NADPH [48]. On the other hand, CA is a substance that can have antioxidant and/or pro-oxidant activity depending on the dose administered, the pH, and its environmental conditions. In addition, the equilibrium state in which it is found with carnosol could modify the results [70]. In this sense, in the tumors, the extracellular pH is usually acidic as a consequence of poor perfusion and high acid production in this microenvironment [71]. Under in vivo conditions, melanoma cells maintain intracellular pH in a viable range for cell survival despite the extracellular tumor pH being normally below 7.0. In our study, CA could benefit from these conditions to increase its effect on tumor cells and, as with alkylating agents and platinum-containing compounds, they could probably be selective, acting better on cells found in acidic tumor beds [71,72]. Obviously, more studies are needed to confirm our results and better analyze the radiosensitizing capacity of CA and its interaction with radiation-induced ROS (using the DPPH assay, iron-reducing antioxidant power assay, nitroblue terazolium reduction (NBT) assay, or ROS assay), along with determination of the capacity of detoxifying enzymes such as SOD and glutathione peroxidases (GPx).

On the other hand, it seems that the trigger of responses in the RIBE is the activation of a signaling pathway sensitive to redox through the mitochondrial induction of ROS/RNS [69,73]. The amplification and propagation of these stress signals causes DNA damage, apoptosis in neighboring tissues, and the activation of a systemic inflammatory response [74]. This signaling cascade produces a variety of effects in recipient cells such as the production of MN, SCE, mutations, decreased cell survival, increased apoptosis, and formation of protein foci associated with the response to DNA damage [73,74]. In our study, we sufficiently determined the genotoxic and cytotoxic effects induced by sera from irradiated animals and by irradiated culture media obtained from irradiated cell cultures on recipient cells, and determined the increase in MN frequency in the three types of assays used.

It has previously been described that the factors that may be involved are elements produced by irradiated cells, namely: cytokines, TGF-β, RNA, TNF-α, IL-6, IL-8, and ROS [74,75,76]. A sustained increase in ROS production can be assumed to contribute to the genotoxic effect by continuously inducing oxidative DNA damage. It has been suggested by different authors that the RIBE mechanism on the genotoxic effect on DNA in RIBE recipient cells could be due to the action of ROS induced by ionizing radiation on recipient cells. In addition, it has been suggested that these ROS could produce a wide variety of effects by modifying signaling pathways [77,78,79,80,81,82].

In our study, we found that the administration of CA lacked cytoprotective effects on the receptor cells due to its inability to inhibit or eliminate the signals that are sent through the serum of the irradiated animals or the culture medium of the irradiated cells in both cell lines. Our results seem to confirm what was described by Harada et al. [81], who described radioinduced free radicals as participating in the formation of signals in cells directly traversed by IR; however, they may not participate in the elimination of bystander signals received by distant irradiated cells.

A few decades ago, Konopackac et al. proposed that interceptors of ROS and in vitro production of peroxides (such as Vitamin C) reduce the frequency of chromosomal breakage and reduce their contribution to MN production in non-bystander cells, thereby protecting them from damage caused by the RIBE effect [79,80]. Recently, using a water-soluble antioxidant substance (rosmarinic acid), we described a reduction of MN frequency in bystander cells, though no cytoprotective effect that increased cell survival in bystander cells. We suggested that rosmarinic acid, unlike vitamin C, does not present significant activity against lipid peroxidation processes that increase IR-induced oxidative DNA damage, which would generally be the case in a RIBE [78,79,80,81,82,83,84]. However, CA is a lipid-soluble substance considered to have a significant capacity to act against lipid peroxidation processes, showing similar results: reduction in the frequency of MN in the bystander effect together with a lack of protection of cell survival in recipient cells. On the other hand, administration of CA to B16F10 melanoma bystander cells (BysCA) produces an increase in the GSH/GSSG ratio compared to untreated cells (Bysi). This increase is similar to that determined in normal PNT2 cells treated with CA, and could be interpreted as an additive/synergistic effect of CA which permits an increase of reduced cellular GSH in the absence of oxidative stress [44]. These results suggest that IR-induced free radicals in directly irradiated cells do not participate in the decrease in cell survival in recipient cells through RIBE.

## 5. Conclusions

CA is an antioxidant compound with significant genoprotective and radioprotective capacity. However, in melanoma cells it acts in a paradoxical way and becomes a radiosensitizing agent, significantly reducing cell survival. Potentially, knowing the mechanisms of action of substances such as CA could help create new applications that allow protecting healthy cells and exclusively damaging neoplastic cells, thus presenting a new desirable strategy for cancer patients who need to undergo radiotherapy.

## Figures and Tables

**Figure 1 antioxidants-11-02166-f001:**
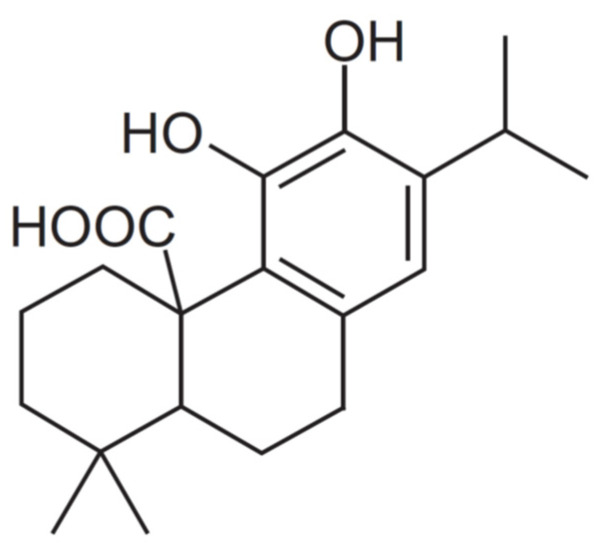
Chemical structure of carnosic acid [1].

**Figure 2 antioxidants-11-02166-f002:**
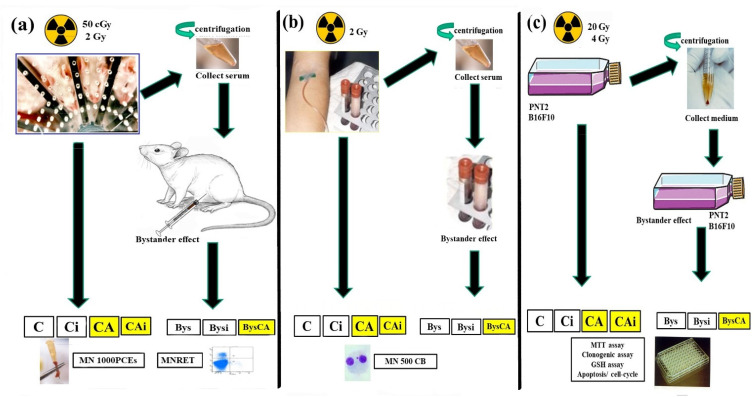
Experimental design of the study: (**a**) In vivo micronucleus assay (in mouse bone marrow (MNPCEs) and reticulocytes (MNRET)); (**b**) In vitro micronucleus assay (MNCB); (**c**) Survival cell assay (MTT cytotoxicity assay, clonogenic assay, GSH assay, and apoptosis (C, control; CA, treated with carnosic acid; Ci, irradiated control; CAi, irradiated previously treated with carnosic acid; Bys, treated with serum/medium non irradiated; Bysi, treated with serum/medium previously irradiated; BysCA, treated with irradiated serum/medium previously treated with carnosic acid).

**Figure 3 antioxidants-11-02166-f003:**
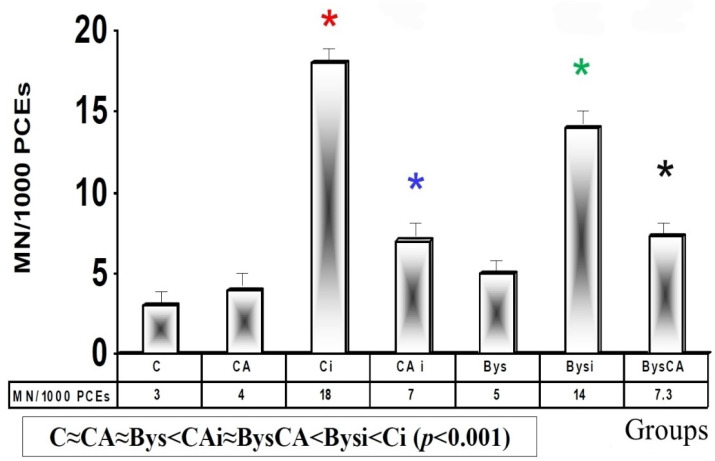
Frequency of micronucleated polychromatic erythrocytes (MNPCEs) in mouse bone marrow (C, control; CA, treated with carnosic acid; Ci, irradiated control; CAi, treated with CA and irradiated; Bys, treated with serum non irradiated; Bysi, treated with serum previously irradiated; BysCA, treated with irradiated serum previously treated with carnosic acid) (*
*p* < 0.001 versus C; *
*p* < 0.001 versus Ci; *
*p* < 0.01 versus Ci; * *p* < 0.01 versus Bysi).

**Figure 4 antioxidants-11-02166-f004:**
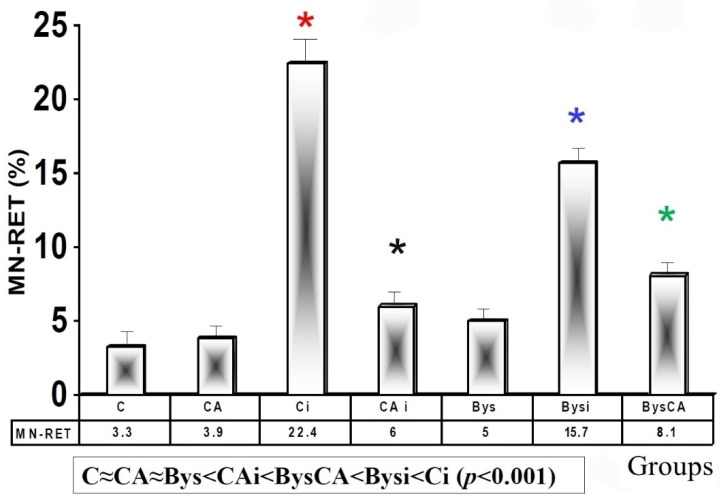
Frequency of reticulocytes in blood determined by flow cytometry 48h after exposure to ionizing radiation (% MN-RET) (C, control; CA, treated with carnosic acid; Ci, irradiated control; CAi, irradiated previously treated with carnosic acid; Bys, treated with serum non irradiated; Bysi, treated with serum previously irradiated; BysCA, treated with irradiated serum previously treated with carnosic acid) (*
*p* < 0.001 versus C; *
*p* < 0.001 versus Ci; *
*p* < 0.01 versus Ci; * *p* < 0.01 versus Bysi).

**Figure 5 antioxidants-11-02166-f005:**
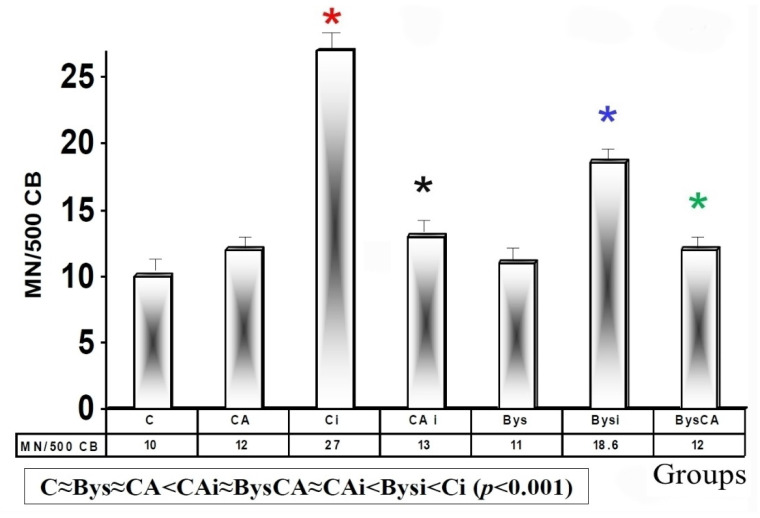
Frequency of micronuclei in cytokinesis-blocked human lymphocytes (CBMN) (C, control; CA, treated with carnosic acid; Ci, irradiated control; CAi, irradiated previously treated with carnosic acid; Bys, treated with serum non irradiated; Bysi, treated with serum previously irradiated; BysCA, treated with irradiated serum previously treated with carnosic acid) (*
*p* < 0.001 versus C; *
*p* < 0.001 versus Ci; * *p* < 0.01 versus Ci; * *p* < 0.01 versus Bysi).

**Figure 6 antioxidants-11-02166-f006:**
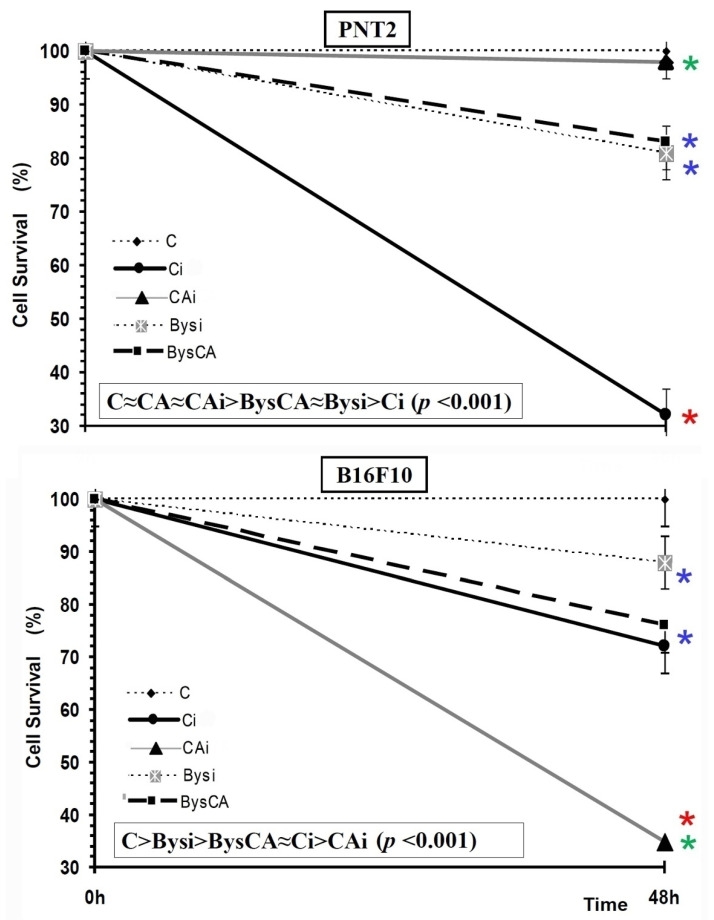
MTT assay: Cell survival of all groups studied in PNT2 and B16F10 (C, control; CA, treated with carnosic acid; Ci, irradiated control; CAi, irradiated previously treated with carnosic acid; Bys, treated with serum non irradiated; Bysi, treated with serum previously irradiated; BysCA, treated with irradiated serum previously treated with carnosic acid) (*
*p* < 0.001 versus C; *
*p* < 0.01 versus C; *
*p* < 0.001 versus Ci). Groups CA and Bys are not represented in the figure, since they coincide with group C. Data are mean ± SE of six independent experiments.

**Figure 7 antioxidants-11-02166-f007:**
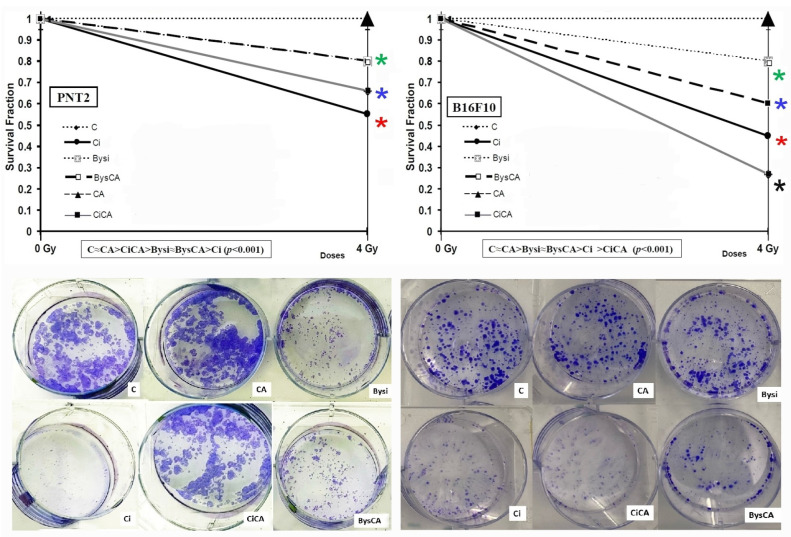
Clonogenic assay: Survival Fraction (SF) of PNT2 and B16F10 cells irradiated with 4 Gy of X-rays (C, control; CA, treated with carnosic acid; Ci, irradiated control; CiCA, irradiated previously treated with carnosic acid; Bys, treated with serum non irradiated; Bysi, treated with serum previously irradiated; BysCA, treated with irradiated serum previously treated with carnosic acid) (*
*p* < 0.001 versus C; *
*p* < 0.01 versus C; *
*p* < 0.01 versus Ci; * *p* < 0.01 Ci). Groups CA and Bys are not represented in the figure, as they coincide with group C. Data are mean ± SE of six independent experiments.

**Figure 8 antioxidants-11-02166-f008:**
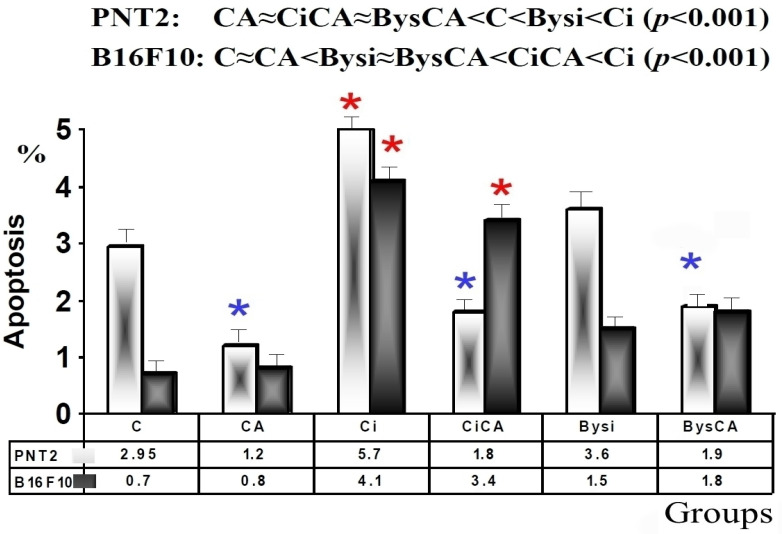
Percentage of PNT2 and B16F10 cells in apoptosis 48h after exposure to 20 Gy of X-rays (C, control; CA, treated with carnosic acid; Ci, irradiated control; CiCA, irradiated previously treated with carnosic acid; Bysi, treated with previously irradiated serum; BysCA, treated with irradiated serum previously treated with carnosic acid) (*
*p* < 0.001 versus C; *
*p* < 0.01 versus C). Data are mean ± SE of six independent experiments.

**Figure 9 antioxidants-11-02166-f009:**
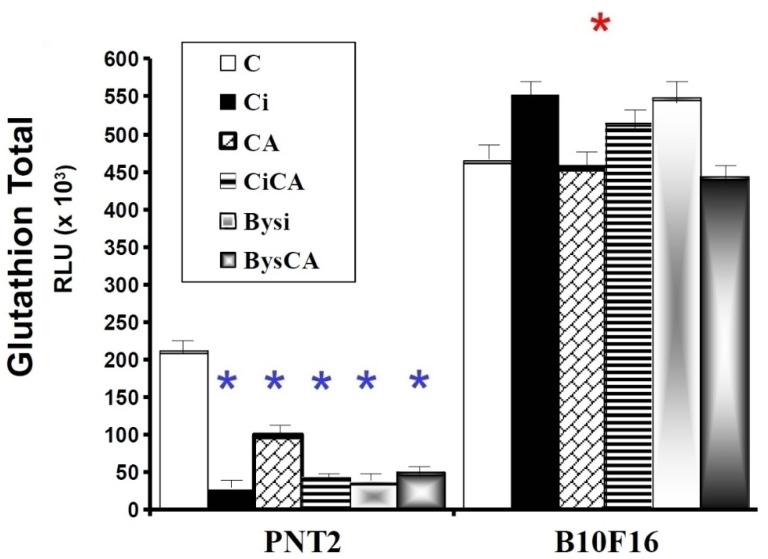
GSH assay: total glutathione concentrations of the different groups studied (C, control; CA, treated with carnosic acid; Ci, irradiated control; CiCA, irradiated previously treated with carnosic acid; Bysi, treated with previously irradiated serum; BysCA, treated with irradiated serum previously treated with carnosic acid) (*
*p* < 0.001 versus PNT2 control (C); *
*p* < 0.001 versus PNT2 control (C)). Data are mean ± SE of six independent experiments.

**Figure 10 antioxidants-11-02166-f010:**
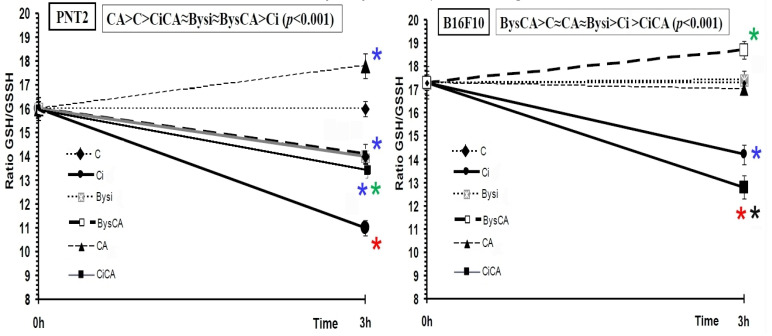
GSH/GSSG ratios of the different groups studied (C, control; CA, treated with carnosic acid; Ci, irradiated control; CiCA, irradiated previously treated with carnosic acid; Bysi, treated with previously irradiated serum; BysCA, treated with irradiated serum previously treated with carnosic acid) (*
*p* < 0.001 versus C; *
*p* < 0.01 versus C; *
*p* < 0.01 versus Ci; * *p* < 0.05 versus Ci). Data are mean ± SE of six independent experiments.

## Data Availability

The data presented in this study are available in article.

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
