# Peer review of "Paradoxical Radiosensitizing Effect of Carnosic Acid on B16F10 Metastatic Melanoma Cells: A New Treatment Strategy"

_antioxidants, 2022, doi:10.3390/antiox11112166_

Round 1

Reviewer 1 Report

The article addresses a topic of interest such as the selective radiosensitization of tumor cells exerted by an antioxidant substance (CA), while exerting a radioprotective effect on healthy cells.

The introduction is brief and direct, it explains the state of the art on the properties of CA and clearly establishes the objectives to be achieved. The methodology to achieve the stated objectives is adequate.

However, the manuscript needs to be improved by minor changes affecting English and spelling errors, and major changes affecting the description of results. In the description of the results, it is necessary to specify to a greater extent the level of the effect of the different treatments and, in some cases, to clarify between which groups the comparative differences are established.

Abstract

The text sometimes creates confusion by commenting out the term cells in isolation. It is recommended to clarify this and say normal cells when referring to the effect on them (as is done in line 59 of the introduction).

Introduction

Line 51. The verb is missing. It should say: Melanoa is….

Line 56. The subject is missing. It should say: In this work, we evaluated…

Results

In order to better estimate the effect of the applied treatments, in all the results where it is not stated, the number of times that cell survival or SF is reduced or increased with respect to the corresponding control must be indicated. This can be put in parentheses, next to the corresponding p-value.

Lines 304-305. Delete “…, which expresses its ability to protect against genotoxic damage induced by ionizing radiation”, because the same idea is repeated in line 307.

Line 324. Where it says “CAi, previously irradiated and treated with carnosic acid”, does it mean: CAi, previously treated with carnosic acid and then irradiated? If so, it should say: “CAi, treated with CA and irradiated” (as it says on line 83).

Lines 370-371. Delete “…, which expresses its ability to protect against genotoxic damage induced by ionizing radiation”, because the same idea is repeated in lines 372-373.

Lines 400-401. It says: “The administration of CA to PNT2 cell cultures before irradiation (CAi) produced a significantly different increase in cell survival (p < 0.001),….”. It must be indicated with respect to which group. From what is indicated in Figure 6, it can be deduced that the increase is with respect to the Ci group, but it must be stated in the text.

Line 404. Delete “.., (Figure 6)”. It is already indicated before and after.

Line 414. The caption mentions the groups CA (treated with carnosic acid) and Bys (treated with serum non irradiated); however, lines corresponding to these groups do not appear on the graph. To avoid confusion, it is recommended to indicate in the figure footer that the CA and Bys groups are not represented, since they coincide with the C group.

Lines 428-430. It says: “However, contrary to expectations, the administration of CA to B16F10 cell cultures before irradiation (CAi) produced decrease in cell survival that was shown to be a statistically significant difference (p < 0.001) expressing an increase in cytotoxic damage induced by X- rays”. It is to be noted that the administration of CA to B16F10 cell cultures before irradiation (CAi) produced a greater reduction in cell survival than irradiation alone; Likewise, the group with which the statistical comparison is made must be mentioned (in this case, it should be said with respect to the Ci group).

Lines 440. It is not clear between which groups the statistical difference of p<0.05 is established.

Lines 442. Check the data of 100.1. Figure 6 seems to indicate a higher value.

Line 449. As in Figure 6, the caption mentions groups CA (treated with carnosic acid) and Bys (treated with serum non irradiated); however, lines corresponding to these groups do not appear on the graph. To avoid confusion, it is recommended to indicate in the figure footer that the CA and Bys groups are not represented, since they coincide with the C group.

Lines 462. Indicate between which groups the statistical difference of p<0.001 is established.

Lines 471. It is not clear between which groups the statistical difference is established.

Line 478. As in figures 6 and 7, the caption mentions groups CA (treated with carnosic acid) and Bys (treated with serum non irradiated); however, lines corresponding to these groups do not appear on the graph. To avoid confusion, it is recommended to indicate in the figure footer that the CA and Bys groups are not represented, since they coincide with the C group.

Line 494. One point is missing before the “All” term.

Lines 499-500. It is said: “…the administration of CA to these cells (BysCA) did not produce a decrease in SF with respect to cells treated with irradiated medium without CA (Bysi), expressing a lack of toxic effect of CA in these control cell cultures” . The effect and p values must be specified, since figure 8 seems to express differences between the BysCA and Bysi groups.

Line 501. Delete the point before the term (Bysi).

Line 503. Delete “(Figure 6)”. It does not correspond to Figure 6, but to Figure 8, and furthermore, it is already indicated later.

Line 538. It says: “(CiCA)”; should say: (CA).

Lines 542-543. It is not clear between which groups the statistical difference is established.

Lines 552-553. In Figure 10, neither a) nor b) appear. However, in the figure caption a) and b) are indicated. This inconsistency needs to be corrected. Furthermore, the pattern of each of the bars is confusing with respect to the description of each group indicated next to each small square. The fourth bar of PTN2 corresponds to CiCA? It does not seem to coincide with the gray square corresponding to CiCA. Improve the choice of the framework of the bars.

On the other hand, no descriptive mention is made in the text of the total glutathione levels, corresponding to the different groups. Only the comparison between the controls of the two cell types is made.

Line 587. There is an inconsistency between the text and figure 11. In my opinion, where it says “in irradiated cells (BysCA)”, it should say treated with previously irradiated serum (Bysi). Clarify this inconsistency. On the other hand, the effect of the group (BysCA)” must be described in the text (at the level of line 598), since there is no mention of the increase produced by CA in said group.

Discussion

Line 614. A point is missing before the In term.

Line 621. Delete one of the “and” terms (it is repeated).

Line 666. Where it says B16F18, should it say B16F10?

Line 621. Delete one of the “cell” terms (it is repeated).

Lines 679-681. Review the paragraph corresponding to these lines to improve your understanding.

In addition:

- It is necessary to discuss why in B16F10 cells the addition of CA to irradiation reduces apoptosis with respect to the irradiated group and, on the other hand, this addition reduces cell survival, increasing the effect of irradiation on said tumor population.

- Discuss why there is an increase in the GSH/GSSG ratio in the BysCA group with respect to the control group

Reviewer 2 Report

The authors present a research work regarding the radioprotective/ radiosensitizing role of Carnosic acid (CA) on cells exposed to X-rays and the radiation induced bystander effect.

Line 34 Please edit to add ref to "Figure 1" since you describe its chemical structure and origin. 

Line 51 the phrase needs edit. Is it proper to say high levels for epidemiological data? Please add some relative references (lines 51-54)

What is the need to perform this study? What is lacking from previous studies (ref 4-10) regarding CA?

How was determined the dose of CA since limited PK data are available?

Line 94 What specialists? Under what terms and how any bias was avoided. 

Line 131 Why supposedly? Informed consent? 

Cell culture. Why only one concentration of CA and under what conditions did the 25μM was chosen?

Since the IR were performed immediately (line 160), the CA's effect was referring to the soluble in the medium CA or the one that is inside the cells?

Figure 2 needs improvement in resolution

Figure 3, 4, 5, 9, 10 maybe you use horizontal brackets to indicate the statistical significance. The experiments are too many and the reader have to check the text continuously. 

Figures 5-6 needs edit (boundaries etc.). 

Figure 8 please improve resolution, especially the plates needs to be bigger and clearer. 

Generally all figures need improvements. Maybe export all graphs as images and then import them to the office template. 

Line 611 ,612. Please provide references. Bioavailability is a very specific term in pharmacokinetics. What the authors mean by absorption? from GI-track? Better edit this phrase and omit terms that have specific meaning in Pharmacokinetics. Alternative you can use synonyms or at least explain what they mean in this case. For example we don't see any results regarding the uptake from the cells of CA. How we ensure its action is due to intracellular concentration and not in the medium. 

What was the level of uptake in the cells of the 25μM CA?

Is it that paradoxical for a compound such as CA (with aromatic structure) to show redox action depending on the environment that is placed? Were both cell lines (as cells) by default in same/similar "oxidation-reduction" status or environment? Maybe the oxidation/reduction ratio for the compound changed in the two cell lines making it act differently. Any comment?

Reviewer 3 Report

Dear Authors,

please address these comments below.

- Line 46: please, cut (CA) that has already been specified above.

- Figure 2: it is not understandable. The figure results confused and unfocused. Please, revise it substantially.

- Lines 302-303: change administration in administered

- Figure 3: the horizontal line "Treatments" below the x-axis is not clear. Please, change it in a more "friendly" format, also in text. Also, add the asterisks when significant.

- Paragraph 3.1.2: I do not understand why it is referred as "in vitro micronuclei assay" . Is this paragraph describing experiments on irradiated animals, as specified in the text? Shouldn't be referred as "in vivo" ?

- Pragraph 3.2: the title is written in Spanish. Please change it.

- Figures 6 and 7: please, combine the paragrahs and these two figures in just one figure, referring to these two cell lines.

- 3.6 Please, change the title in "Apoptosis assay"

- Figure 10: there is a mistake in the histogram legend. Please, revise the color code.

Round 2

Reviewer 1 Report

The manuscript needs to be improved by minor changes affecting English and spelling errors, and major changes affecting the description of results.

Introduction

Line 54-56 (version 2). it is said: “For this reason, it is extremely important development of novel therapeutic arsenals targeted at improving the survival of affected patients is warranted [11-12]”.

The grammatical structure of the paragraph needs to be improved. For example:

“For this reason, it is extremely important to develop new therapeutic arsenals aimed at improving the survival of affected patients”.

Results

Taking into account previous comments regarding the first version, the following changes have not been made:

-          Lines 499-500 (version 1). It is said: “…the administration of CA to these cells (BysCA) did not produce a decrease in SF with respect to cells treated with irradiated medium without CA (Bysi), expressing a lack of toxic effect of CA in these control cell cultures” . The effect and p values must be specified, since figure 8 seems to express differences between the BysCA and Bysi groups.

The authors indicate that they have carried out the modifications corresponding to this aspect, but this is not the case.

-          Line 538 (version 1).

In the previous review, I commented the following:

It says: “(CiCA)”; should say: (CA).

The authors indicate that they have carried out the modifications corresponding to this aspect, but this is not the case.

Lines 547-550 (version 1).

On the other hand, no descriptive mention is made in the text of the total glutathione levels, corresponding to the different groups. Only the comparison between the controls of the two cell types is made.

The authors indicate that they have carried out the modifications corresponding to this aspect, but this is not the case.

In this sense, it is noteworthy that CA treatment in PNT2 cells decreases total GSH levels with respect to control cells. This is so? Check and describe in the text the effect of the different treatments on total glutathione levels.

Discussion

Taking into account previous comments regarding the first version, the following changes have not been made:

-          Line 614 (version 1). A point is missing before the In term (after reference 13).

The authors indicate that they have carried out the modifications corresponding to this aspect, but this is not the case.

-          Line 666 (version 1). Where it says B16F18, should it say B16F10?

The authors indicate that they have carried out the modifications corresponding to this aspect, but this is not the case.

-          Lines 679-681 (version 1). Review the paragraph corresponding to these lines to improve your understanding.

The authors indicate that they have carried out the modifications corresponding to this aspect, but this is not the case.

On the other hand, it would be advisable to discuss why CA treatment in PNT2 cells decreases total GSH levels compared to control PNT2 cells.

Reviewer 2 Report

I would like to thank the authors for revising their manuscript according to the suggestions. The manuscript is further improved. There are some issues that should be further addressed within the text. Some comments follow: 

"Melanoma is a cancer responsible for the highest mortality rate among patients with skin cancer and with a high incidence rate in the white population."

Please (according also to the initial round) provide relative references for this statement adding also any differences among different races.  The references that you provided are not covering or referring to this statement. 

Specialists. Are they within the study group? The authors response with three different expertise. Who is more capable to evaluate it? Please also add this sentence to the text and explain how any bias was avoided. What double-blind means in this case.

Cell culture

Please provide a supplementary file with the other series of experiments explaining how you end up with these parameters that are presented within the main text. 

Figure 7 still needs improvement in resolution maybe use a seperate figure for the plates

I understand the response regarding the term "Bioavailability". But it is not proper to be used. See for example https://doi.org/10.1016/B978-0-12-386454-3.00419-X.

Bioavailability (F) is a term used to describe the percentage (or the fraction (F)) of an administered dose of a xenobiotic that reaches the systemic circulation. Bioavailability is practically 100% (F = 1) following an intravenous administration. Bioavailability could be lower (F ≤ 1) and in some cases almost negligible for other routes (e.g., oral and dermal), depending on how efficiently a xenobiotic crosses various biological membranes (e.g., skin and stomach). Additionally, whether or not tissues or organs (e.g., skin and liver), through which xenobiotics pass before reaching the systemic circulation, are capable of metabolizing the substance. The latter phenomenon is known as a first-pass effect. Bioavailability may vary considerably between xenobiotics or even between batches of a given xenobiotic. For example, therapeutic drugs must undergo bioavailability testing to ensure reliable dosing throughout treatment. The blood concentration of the administered drug is used as an index of bioavailability.

Did the authors studied this phenomenon? I think not. They even state this in their response. So, even if the authors are using terminologies that were described in other works I suggest to remove them and describe it with a synonym word. In the context of a preliminary, preclinical study as this one, the terminology should be specific for the reader. 

Please include the last comment of the oxidant/redox action depending on the environment within the discussion section commenting the environment that melanoma is creating in vivo. See some relative refs. 

Wahl ML, Owen JA, Burd R, Herlands RA, Nogami SS, Rodeck U, Berd D, Leeper DB, Owen CS. Regulation of intracellular pH in human melanoma: potential therapeutic implications. Mol Cancer Ther. 2002 Jun;1(8):617-28. PMID: 12479222.

Martínez-Zaguilán R, Seftor EA, Seftor RE, Chu YW, Gillies RJ, Hendrix MJ. Acidic pH enhances the invasive behavior of human melanoma cells. Clin Exp Metastasis. 1996 Mar;14(2):176-86. doi: 10.1007/BF00121214. PMID: 8605731.

Reviewer 3 Report

Dear Authors, 

I read all the modifications and improvements made to your manuscript.

I see that the authors tried to improve the quality of the content and figures, as suggested by the reviewers. However, I think that the manuscript still presents flaws. For instance, Figure 2 is still of low quality and the content is not clear. Some results are not sound. I read all the coments of the other Reviewers concerning the Results and the Discussion. I completely agree.

Round 3

Reviewer 2 Report

I would like to thank the authors for providing an updated version of their work addressing the comments made during the review process. The manuscript can be accepted and be further processed for publication. Please edit line 233 p4 simples-> samples. 

Reviewer 3 Report

I saw the improved version of Figure 2.